# Reduced Breast and Ovarian Cancer Through Targeted Genetic Testing: Estimates Using the NEEMO Microsimulation Model

**DOI:** 10.3390/cancers16244165

**Published:** 2024-12-13

**Authors:** Lara Petelin, Michelle Cunich, Pietro Procopio, Deborah Schofield, Lisa Devereux, Carolyn Nickson, Paul A. James, Ian G. Campbell, Alison H. Trainer

**Affiliations:** 1Parkville Familial Cancer Centre, Peter MacCallum Cancer Centre, Melbourne 3052, Australiaalison.trainer@petermac.org (A.H.T.); 2Parkville Familial Cancer Centre, The Royal Melbourne Hospital, Melbourne 3052, Australia; 3The Daffodil Centre, a Joint Venture Between Cancer Council NSW and the University of Sydney, Sydney 2011, Australia; 4Melbourne School of Population and Global Health, University of Melbourne, Melbourne 3052, Australia; 5Charles Perkins Centre, Faculty of Medicine and Health, The University of Sydney, Sydney 2006, Australia; michelle.cunich@sydney.edu.au; 6Sydney Local Health District, Sydney 2050, Australia; 7Sydney Institute for Women, Children and Their Families, Sydney Local Health District, Sydney 2050, Australia; 8Implementation and Policy, Cardiovascular Initiative, The University of Sydney, Sydney 2006, Australia; 9Centre for Economic Impacts of Genomic Medicine (GenIMPACT), Macquarie Business School, Macquarie University, Sydney 2113, Australia; 10Research Division, Peter MacCallum Cancer Centre, Melbourne 3052, Australia; 11Sir Peter MacCallum Department of Oncology, University of Melbourne, Melbourne 3052, Australia; 12Department of Medicine, University of Melbourne, Melbourne 3052, Australia

**Keywords:** genetic testing, predictive testing, microsimulation, hereditary breast and ovarian cancer

## Abstract

Genetic testing for hereditary breast and ovarian cancer can identify individuals at an increased risk of cancer who could benefit from intensive cancer screening and prevention strategies. Our aim was to develop a simulation model that accurately reflects genetic testing and cancer risk management practices in Australia by incorporating multi-generational family units, real-world adherence and uptake of clinical recommendations. This study found that expanding genetic testing to all newly diagnosed breast cancer patients reduced population cancer incidence but did not lead to an increase in life-years saved when compared to optimising targeted testing of patients eligible under the current criteria. This model is a novel platform for evaluating the clinical and cost-effectiveness of genetic testing interventions for hereditary breast and ovarian cancer.

## 1. Introduction

Germline genetic testing for hereditary breast and ovarian cancer (HBOC) enables targeted treatment and risk management interventions for women who carry pathogenic or likely pathogenic (P/LP) variants in a cancer predisposition gene, such as *BRCA1*, *BRCA2* or *PALB2*. Between 2% and 7% of women diagnosed with breast cancer carry a germline HBOC P/LP variant, with a higher prevalence in young-onset cases and triple-negative tumours [1,2]. The prevalence of high-risk germline P/LP variants in epithelial ovarian cancer ranges from 10% to 18% and is predominately associated with the high-grade serous subtype [3,4].

Most genetic testing guidelines recommend germline testing when there is both a personal history of breast and/or ovarian cancer and other characteristics suggestive of a high-risk P/LP variant [5,6,7]. In Australia, a pre-test probability of 10% is required to qualify for publicly funded HBOC testing [8]. Germline testing is well accepted among patients with breast and ovarian cancers, with an uptake rate of 95–97% [9,10,11]. However, current testing guidelines have low sensitivity for identifying all carriers, substantively due to suboptimal patient referral rates and narrow eligibility criteria [12,13,14]. Approximately 50% of P/LP variant carriers do not present with a family history significant enough to trigger a referral to genetics services [3,15], and testing guidelines can be difficult for non-genetics specialists to navigate [16]. 

P/LP variant status in women with cancer can influence treatment decisions, such as opting for mastectomy over breast-conserving surgery and increasing access to targeted or systemic therapy [17,18,19]. P/LP variant carriers can mitigate their future cancer risk by undergoing annual breast screening at a young age for early detection and/or medication for cancer prevention or through risk-reducing surgery [20]. First-degree relatives of known carriers have a 50% risk of harbouring the familial variant, which is significantly higher than the 10% threshold required for testing. Thus, the ability to identify and test unaffected relatives of known P/LP variant carriers, known as predictive testing, is a significant component in determining the optimal genetic testing practice.

Existing evaluations of germline HBOC testing strategies have used simulation models to estimate the effectiveness and cost-effectiveness of expanded testing criteria. Such modelling has included testing all women with breast cancer, all women with ovarian cancer, and population-wide testing [21,22,23]. Depending on the genetic testing criteria evaluated, clinical and cost-effectiveness estimates can be highly sensitive to the number of unaffected relatives who undergo predictive testing [24]. A significant limitation of previous simulation models is that rather than directly modelling relatives, they made simplifying assumptions regarding the number of at-risk unaffected family members, how many predictive tests would be expected, and subsequent uptake of cancer risk management in those who test positive [25]. Prior modelling studies also do not consider familial clustering of cancers due to inherited common genomic variants (polygenic risk). These are important issues as accurate modelling of current practice provides an important benchmark against which proposed changes to genetic testing guidelines can be assessed.

Our objective was to develop a novel and flexible microsimulation model that reflects the current practice for HBOC genetic testing in Australia and directly incorporates variation in family size, the dynamic nature of predictive testing within families, and likely uptake and adherence to cancer risk management recommendations. The model, called NEEMO, was designed to enable the evaluation of different HBOC genetic testing approaches. Here, we describe the model design, validation, and practical application of the model in projecting long-term clinical outcomes for four HBOC testing scenarios. 

## 2. Materials and Methods

### 2.1. Model Overview

The populatioN gEnEtic testing MOdel (NEEMO) is a microsimulation model that simulates the life histories of breast and ovarian cancers in a subset of women in the Australian population as individuals are structured within biological family units. It was coded in Python version 3.9 (Python Software Foundation, Wilmington, DE, USA). Model development was informed by the literature, data from an established and large Australian familial cancer service, and a population-based genetic testing research study of women enrolled in the Lifepool cohort [15].

NEEMO comprises two main components:Part 1: Model population generation (the “input population”, see Appendix A; andPart 2: The main simulation (referred to as the simulation; see Appendix A).

### 2.2. Population Generation

#### 2.2.1. Overview

The population is constructed by initialising the desired number of index women, called probands, to serve as the centre of each family unit. All probands have no personal history of cancer and can be specified anywhere between 20 and 59 years of age on entry into the simulation. Male and female first- and second-degree relatives are generated for each proband. The distributions of family structure and age at commencement of the main simulation (Part 2 of NEEMO, also referred to as age at simulation entry) are based on data obtained from 11,084 unique family pedigrees from a genetics clinic in Melbourne, Australia.

First-degree relatives can include parents, children, and siblings of the probands. Second-degree relatives can be grandparents, aunts and uncles (piblings), nieces and nephews (niblings), and grandchildren. All relatives in the input population are assigned an age at simulation entry, vital status (alive, deceased, not yet born), HBOC genotype (see below for details), and a personal breast and/or ovarian cancer history. 

It is assumed that each proband has two parents and four grandparents. For all other relatives (siblings, piblings, children, grandchildren, and niblings), the number of relatives is based on age-specific cumulative distribution functions for parity (number of children). Further details regarding population generation are provided in Appendix A, including parity estimates and birth age (Appendix A). 

#### 2.2.2. Population Genotypes

The genotype includes both the presence and absence of a single P/LP variant (monogenic risk) in genes *BRCA1*, *BRCA2*, *PALB2*, *CHEK2* (1100delC only), *ATM*, *RAD51C*, *RAD51D*, or *BRIP1*, and a breast and ovarian cancer-specific polygenic component. The prevalence of P/LP variants was sourced from published estimates of the Australian Lifepool study cohort, which is a population-based cohort of women. These women were recruited through BreastScreen Victoria (a population-based breast screening programme) and were unaffected by cancer at enrolment (Appendix A) [15]. Expanded HBOC genes associated with additional non-breast/ovarian cancers (such as *TP53* and *CDH1*) were excluded due to complexities around competing comorbidities and mortality [7]. Since monogenic P/LP variants explain only a proportion of the familial clustering for breast/ovarian cancer, an inherited polygenic component was incorporated into the model. The inheritance of monogenic and polygenic risks is described in the Appendix A.

### 2.3. Main Simulation Model Structure

The main simulation starts with the defined input population and then models events moving forward in time in yearly cycles. The model includes modules for (i) breast and ovarian cancer natural history, (ii) cancer-specific and other-cause mortality, and (iii) cancer history and genetic testing within families, as well as the following clinical events:Genetic testing, being either:Full P/LP variant detection: genetic sequencing and large genomic rearrangements, either with a high-risk gene panel (*BRCA1*, *BRCA2* and *PALB2*), or an extended gene panel (high-risk genes, plus *ATM*, *BRIP1*, *CHEK2*, *RAD51C*, *RAD51D*); orPredictive genetic testing: Targeted sequencing for a known family P/LP variant;Breast cancer screening: digital mammography and/or breast magnetic resonance imaging (MRI);Risk-reducing surgeries: bilateral risk-reducing mastectomy (BRRM), contralateral risk-reducing mastectomy (CRRM), and risk-reducing salpingo-oophorectomy (RRSO).

NEEMO includes real-world adherence and uptake of cancer risk management strategies, as this was previously shown to impact comparative effectiveness and cost-effectiveness estimates [26,27,28]. The simulation can be specified to use a time horizon of anywhere between 1 and 160 years (cycles). The input population can be specified to be either static or dynamic, defined as follows: for static populations, no new individuals can enter the simulation once it has begun (for example, future grandchildren not yet born at the start of the simulation are not modelled). Alternatively, for dynamic populations, new family members can be born as the simulation moves forward in time. The sequence within each time cycle is shown in Appendix A. 

#### 2.3.1. Cancer Risk and Natural History

Lifetime risks for breast and ovarian cancer for non-carriers are specified using Australian population cancer incidence data (Appendix A). Gene-specific cancer risk for P/LP variant carriers were estimated from various international cohort studies (Appendix A). Cancer risks for carriers and non-carriers are adjusted according to randomly sampled Z-scores for polygenic risk and published estimates of the hazard ratio per one standard deviation (see Appendix A) [29,30]. Cancer risk is not included for male relatives (for example, *BRCA*-associated male breast and prostate cancer). 

#### 2.3.2. Breast Cancer

For breast cancer, cancer onset is defined in NEEMO as the onset of asymptomatic but screen-detectable disease (preclinical breast cancer). Preclinical cancers are assigned hormone receptor status, tumour grade, and preclinical sojourn time based on logistic and multinomial regression modelling of data from the Surveillance, Epidemiology, and End Results (SEER) Program and a previous analysis of Australian *BRCA1* and *BRCA2* carriers (Appendix A) [27,31]. Within the natural history model, breast cancers are assumed to become clinically diagnosed as symptomatic cancer if the preclinical sojourn time expires prior to the cancer being screen-detected or to the woman undergoing risk-reducing surgery. 

Breast cancers are assigned additional prognostic characteristics on detection, specifically tumour size, lymph node status, and presence of metastatic disease (Appendix A). These characteristics are considered fixed and do not change over the course of the disease once diagnosed. Further description of methods relating to gene-specific tumour pathology and breast cancer characteristics are provided in the Appendix A. 

#### 2.3.3. Ovarian Cancer

For the purposes of this study, ovarian cancer refers to epithelial cancer of the ovary, fallopian tube or peritoneum. No preclinical phase is modelled due to the uncertainty around ovarian cancer onset and growth, and ovarian cancer screening has not been proven to affect either stage at diagnosis or survival [32,33]. As a simplifying assumption, all ovarian cancers diagnosed after the age of 25 years in women with a P/LP variant are assumed to be high-grade serous [3]. For non-carrier ovarian cancers, tumour characteristics are drawn from an analysis of SEER Program data (Appendix A) [31]. Briefly, women are assigned histology of clear cell, endometrioid, serous, mucinous, or “other” type, probabilistically conditional on age at diagnosis. Ovarian cancer grade (1, 2 or 3) is specified as being dependent on histology type and age at diagnosis. Cancers are staged as local, regional or distant. Cancers are more likely to be diagnosed at an advanced stage if they are of a higher grade and/or have a serous histology.

#### 2.3.4. Mortality

Age at death from other causes is based on Australian life tables for both males and females [34]. Female lifetables were adjusted to exclude deaths attributable to breast or ovarian cancer. Separate breast and ovarian cancer survival rates were based on SEER case listing data from women diagnosed from 2006 onwards, to be inclusive of more recent therapies such as trastuzumab for HER2 overexpressing tumours (Appendix A) [31]. 

#### 2.3.5. Clinical Interventions

Three clinical interventions are modelled: (1) clinical genetic services, (2) breast screening and surveillance, and (3) risk-reducing surgery. 

#### 2.3.6. Clinical Genetics Services

Genetic testing includes clinically indicated full P/LP variant detection (diagnostic genetic testing) and predictive testing of relatives of known carriers. A genetic testing module was developed to reflect the current standard of care in Australian clinical practice.

The model specifies that women can only be referred to clinical genetics services for diagnostic genetic testing in the same year that they are diagnosed with breast or ovarian cancer. Women who develop multiple primary cancers during their lifetime can be referred at the time of each new diagnosis. The probability of referral was based on expert opinion and observed referrals to clinical genetics services for triple-negative breast cancer cases and high-grade serous ovarian cancer in Victoria to publicly funded familial cancer services (unpublished observations, Table 1). Based on patterns in the observed data, the probability of referral for women with ovarian cancer is higher for women diagnosed at a younger age (aged <60 years), higher grade, prior history of breast cancer, and non-mucinous histology. Similarly, women with breast cancer are assigned a higher probability of referral if they have a higher grade and young-onset disease. 

Eligibility for publicly funded HBOC genetic testing is based on current practice, meaning that not all women referred to genetics services within the model undergo genetic testing. The Australian Medicare Benefits Schedule (MBS) specifies that a woman is eligible for genetic testing if she is affected by breast and/or ovarian cancer and assessed to be at a 10% risk of carrying a P/LP variant in *BRCA1* or *BRCA2* [8]. In NEEMO, the 10% threshold is assumed to be met for (1) all triple-negative breast cancer cases aged 50 years or younger or of any age with at least one breast or ovarian cancer-affected family member, (2) women with a strong family history (at least three affected first- or second-degree relatives), (3) bilateral breast cancer, (4) all grade 2/3 serous ovarian cancers irrespective of age at diagnosis, and (5) non-mucinous grade 2/3 ovarian cancers aged under 70 years. Privately funded genetic testing is not included in the model.

Within NEEMO, male and female relatives of a known carrier are eligible for predictive testing, provided they are not obligate non-carriers based on previous testing within the family and have not previously undergone full HBOC panel P/LP variant detection. Predictive testing of family members can affect the eligibility of other individuals in the family; for example, if a maternal grandfather returns a positive result, the paternal side of the family is no longer eligible for predictive testing (for assumptions related to predictive testing, see Appendix A). A woman who returns a negative result from a predictive genetic test for a family P/LP variant but later develops cancer is still eligible for the full HBOC panel for diagnostic genetic testing. 

Modelled uptake rates for HBOC testing for cancer-affected women are drawn from Australian data [10,11], and uptake of predictive genetic testing from an analysis of HBOC families at an Australian familial cancer centre. Uptake probability in the model is computed for each cycle for eligible relatives, accounting for (1) sex, (2) current age, (3) degree of relation, and (4) the time elapsed since the P/LP variant was identified in the family (Table 2). 

#### 2.3.7. Breast Screening

Breast screening modalities include digital mammography and breast MRI. Modelled age-specific sensitivity and specificity for mammography alone were drawn from Australia’s population breast screening programme (BreastScreen Australia), which targets women aged 50 to 74 years for two-yearly digital mammograms [35]. For MRI alone or in combination with mammography and mammography in women aged <40 years, sensitivity and specificity estimates are drawn from the literature (Appendix A) [36].

For women without a personal history of breast cancer and who are not known to be at an increased risk, breast screening is modelled through participation in the BreastScreen Australia program. The modelled age at the first screening for the population breast screening programme is based on the observed total participation and rescreening rates (Appendix A) [35]. Approximately 30% of modelled women never participate in BreastScreen. Simulated women can commence participation in BreastScreen at any age between 40 and 64 years, inclusive. The time between screens for BreastScreen participants is based on a cumulative distribution calibrated as part of an existing Australian population-based breast cancer screening model [37].

Eligibility for high-risk screening is specified in NEEMO based on genetic testing results. Once a woman is identified through genetic testing as being at a moderate- or high risk of breast cancer, she is eligible for annual mammography (from age 35–59 years for moderate-risk, 25–64 years for high-risk), and annual MRI (from age 25–59 years for high-risk only). The age at the first high-risk screen is based on a prior study on the uptake in *BRCA* carriers [27]. No data were available for uptake rates by age in moderate-risk women; therefore, as a simplifying assumption, they are assumed to have the same rate of uptake for age at first screen as high-risk women but delayed by an additional ten years for mammography. Women attending breast MRI are assumed to be participating in screening through a specialised high-risk management clinic. The time between screens is simulated in the same way as BreastScreen participants, but screening adherence is instead informed by *BRCA*-specific data [27].

#### 2.3.8. Risk-Reducing Surgery

In NEEMO, known carriers of high-risk P/LP variants are eligible for BRRM and/or RRSO. Estimates for cancer risk reduction following surgery are obtained from systematic reviews and meta-analyses (Appendix A) [38,39]. All women diagnosed with unilateral breast cancer are eligible for CRRM; however, uptake is considerably higher in P/LP variant carriers (Appendix A). Uptake rates for BRRM and RRSO are age- and gene-specific, based on a previous analysis of *BRCA* carriers [27]. Observed uptake rates were available for *BRCA* carriers only, so BRRM uptake estimates for women identified with additional genes (*PALB2*, *ATM*, *BRIP1*, *CHEK2*, *RAD51C*, *RAD51D*) were based on the following assumptions: *PALB2* carriers at same rate as *BRCA2*, *ATM* and *CHEK2* at half the rate of *BRCA2*, and all other carriers are ineligible for BRRM. For RRSO, a similar approach was taken, with uptake rates for *PALB2*, *RAD51C* and *RAD51D* carriers similar to *BRCA2* carriers but only from age 45 years onward for *RAD51C/D* and age 50 years for *PALB2*. *BRIP1* carriers are assumed to be eligible for RRSO from the age of 60 years. 

### 2.4. Model Validation

The simulated family structure for the input population was compared with the general Australian population, reasoning that using pedigrees from people attending a genetics clinic could potentially skew the distribution to larger families, as many individuals attending the clinic were ascertained based on strong known family histories. Target outputs included parity by age group, distribution of mothers’ age at birth, and prevalence of a family history of breast cancer in non-carriers. The underlying prevalence of P/LP variants was validated by checking the resulting prevalence of cancer cases diagnosed during the simulation by cancer subtype. 

Internal validation was performed to confirm that the model output was consistent with the input data. Data from international cohorts were utilised for many of the model inputs due to limitations in accessing sufficient numbers for individual patient data in Australia, in particular for the natural history model. As the model aimed to project outcomes in Australia, the outputs for various elements were externally validated against the summary Australian data available through government reports [40]. Essential targets for external validation included (1) breast cancer survival by age, (2) ovarian cancer survival by age and histology, (3) cancer pathology, (4) genetic testing referral rates, and (5) predictive genetic testing rates. 

### 2.5. Model Outcomes

#### 2.5.1. Scenarios

NEEMO was applied to four genetic testing scenarios and related clinical risk management interventions to assess comparative clinical outcomes. The scenarios evaluated were as follows:No genetic testing: no testing for any HBOC-related P/LP gene variants;Current practice: HBOC genetic testing with a high-risk gene panel (*BRCA1*, *BRCA2*, *PALB2*) based on MBS criteria and current breast and ovarian cancer referral rates;Optimised referral of breast and ovarian cancer: As for (2), with referral of all breast and ovarian cancer cases who are eligible for testing to genetic services;Genetic testing for all breast cancers: As for (3), with the addition of genetic testing being offered to all breast cancer patients aged younger than 80 years.

Clinical pathways to genetic testing under scenario 2 included referral to a clinical genetics service, with subsequent genetic testing for newly diagnosed breast and ovarian cancer cases. Under scenarios 2 and 3, genetic testing eligibility was determined as described earlier. Optimised referral was defined as 100% referral of cancer patients eligible for genetic testing, meaning that all eligible cancer-affected women would undergo HBOC full panel testing. Predictive testing for relatives was included in all scenarios, except scenario 1. While many genetics clinics offer testing for additional genes, such as *CHEK2*, using an extended gene panel test, these were excluded from all scenarios, as they are not currently included in the MBS. Access to population breast screening was active in all scenarios. For all scenarios, only women who were identified as carrying a P/LP variant underwent additional cancer risk management interventions; consequently, no high-risk management was performed in scenario 1. Uptake rates for genetic testing and cancer risk management were the same for scenarios 2–4 (see Table 1 and Table 2 and Appendix A). 

Outcomes included (1) the number of genetic tests performed, (2) the number of relatives at risk of carrying a P/LP variant per proband carrier, (3) detection rates for P/LP variants, (4) uptake of risk-reducing surgeries, (5) breast/ovarian cancer incidence in relatives, (6) life expectancy, and (7) life-years saved. 

#### 2.5.2. Modelled Population

The input population used for the scenarios modelled comprised 1 million female probands aged between 20 and 24 years at entry and all their first- and second-degree relatives (male and female). Probands were specified to be unaffected by cancer at model entry. All family members were followed up until the age of 100 years or death, whichever came first. Importantly, the changes to referral and/or genetic testing for each scenario were applied to all individuals simulated, not only the probands, meaning that it was possible for a family P/LP variant to first be identified in a cancer-affected relative instead of the proband—an advancement on the currently published models. 

A subset of the population was used to evaluate the scenarios, namely, probands who developed breast cancer over the course of the simulation, who were aged under 80 years at diagnosis, and their relatives (for each scenario 1–4). Probands were excluded if they had a P/LP variant detected prior to developing breast cancer. 

#### 2.5.3. Sensitivity Analysis

To allow comparison with a prior study [22], a sensitivity analysis was performed that (1) removed background full P/LP variant detection (diagnostic genetic testing) in relatives so that the family variant could only be first identified in probands and (2) assumed 100% uptake of predictive testing within the same year of a proband’s variant being detected. The sensitivity analysis was limited to scenarios 3 and 4 (optimised referral and testing of all breast cancers), as these most closely reflected the comparator and intervention in the prior study.

## 3. Results

### 3.1. Validation

#### 3.1.1. Population Validation

Simulated family units with a proband aged between 50 and 54 years had an average family size of 22.3 individuals (including deceased), with 50.7% female family members. The distribution of parity by age group (mothers only) closely followed that reported for the general Australian population (Appendix A) [41]. There was a slight skew to lower parity in older ages in the simulated population compared to the observed data, consistent with limits placed on the maximum possible number of children in the model specifications. Maternal age at the time of birth for each child (from first- to fourth-born child) followed the overall pattern observed in the Australian population over four time periods (Appendix A). While the model takes account of age-specific fertility, it does not account for reduced population fertility over time, meaning it simulates only the general trend and is not specific to the calendar year.

Approximately 14.5% of probands without a germline P/LP variant had a family history of at least one first-degree relative with breast cancer, which is in line with reported estimates of between 12 and 16% (Appendix A) [42]. The prevalence of P/LP variants in triple-negative breast cancer cases (9.3%) and non-mucinous ovarian cancer cases (15.9%) was consistent with published studies on unselected cancer cases (Appendix A) [3,43,44].

#### 3.1.2. Cancer Outcomes Validation

The validation results for cancer-related outputs are provided in Appendix A. The model accurately reproduced the expected breast and ovarian cancer incidence for carriers and non-carriers (Appendix A). Ovarian cancer pathology was similar to that reported in Australian government reports, except for the case of histology in women aged 70 years and above at diagnosis, where there were fewer “other” histotypes and increased serous histotypes compared to the observed data (Appendix A). This difference was a result of an intentional change in model inputs for older women following expert advice, as described in the supplementary methods (Appendix A).

For breast cancer, the model slightly under-predicted the number of larger tumours (21–50 mm) detected outside screening, as well as the number of screen-detected lymph node-positive cases (Appendix A). 

Relative survival after ovarian cancer diagnosis was similar for simulated and observed outcomes by histotype, except for serous and other histologies (Appendix A). This difference was expected due to the reclassification of “other” histotypes into serous histotypes for older women, resulting in worse survival for serous cases but improved survival for “other” histologies due to the underlying change in age distribution. The observed relative survival rates were similar to the simulated results when split into 5-year age groups for both ovarian and breast cancer (Appendix A). 

### 3.2. Estimated Genetic Testing Rates and Clinical Outcomes

#### 3.2.1. Genetic Testing Rates

Genetic testing rates were validated using modelled outcomes from current practice (scenario 2). In women diagnosed with breast cancer, subtype-specific modelled referral rates ranged from 11% for HER2 overexpressing tumours to 51% for women with triple-negative tumours (Appendix A). Referrals for triple-negative breast cancer for women aged 40 years or younger at diagnosis increased to 88% (95% CI 69–97%), consistent with published referral rates of 84% (95% CI 79–87%) (Appendix A) [45]. Between 15% and 62% of the women diagnosed with ovarian cancer were referred for genetic assessment (Appendix A). Simulated referral for serous ovarian cancer corresponded with the observed referral rates (Appendix A) [11].

For uptake of predictive testing, an estimated 21% of at-risk relatives completed a genetic test within the first five years of the family P/LP variant being identified, with uptake continuing to increase over time (Figure 1). This equated to an average of 2 relatives being tested per proband in the first 5 years, compared to 1.5 for families seen through a single familial cancer clinic and 1.98 for participants in the Lifepool study and their relatives (Appendix A). Simulated uptake of BRRM and RRSO in unaffected *BRCA1* and *BRCA2* was in line with uptake observed in a familial cancer clinic population and the Lifepool cohort (Appendix A).

#### 3.2.2. Genetic Testing Outcomes

One million probands were simulated for each of the four genetic testing scenarios (Section 2.5.1), with approximately 120,000 of these probands diagnosed with breast cancer over their lifetime. Genetic testing outcomes are reported for probands with breast cancer, as described in Section 2.5.2, and their first- and second-degree relatives. Table 3 describes the underlying prevalence of P/LP variants in these probands and the genetic testing rates for both probands and their relatives. Increased genetic testing rates were modelled to be implemented across the entire simulated population and not confined to probands, meaning the underlying prevalence of P/LP variants in cancer-affected probands decreased with each intervention from 3.53% for the no-testing scenario to 2.39% for testing all breast cancers.

Under scenario 2 (current practice), 9279/120,700 (7.69%) of probands underwent diagnostic genetic testing following their breast cancer diagnosis. This increased to 13.32% under the optimised referral scenario and 99.99% for scenario 4, testing all breast cancers. There was a small number of probands (0.44–0.60%) who had a predictive test for a known family variant in all scenarios except for scenario 1 (no genetic testing). Each expansion of genetic testing led to an incremental decrease in secondary ovarian cancer incidence in probands with a P/LP variant, from 557/4245 (13.12%) with no genetic testing to 49/2867 (1.71%) with all breast cancers tested.

Probands with a P/LP variant had an average of 15.12–15.63 male and female relatives alive (including cancer-affected) at the time of the proband’s breast cancer diagnosis. There was a substantial increase in both diagnostic sequencing and predictive testing performed on relatives in each expanded testing scenario (Table 3).

The estimated mean number of at-risk female relatives with a P/LP variant who were unaffected by cancer at the time of the proband’s diagnosis ranged from 2.04–2.08 per proband carrier (including relatives aged under 18 years). Probands with *PALB2* P/LP variants had more at-risk relatives on average at 2.10–2.22, compared to proband *BRCA1* carriers at 1.74–1.90 relatives per proband and *BRCA2* at 1.70–1.87. The estimated age distribution of unaffected female relatives tended to become slightly older in each scenario as genetic testing expanded. Further details regarding genetic outcomes by age group are provided in Appendix A. 

Current practice detected 39% of probands with a P/LP variant following their breast cancer diagnosis, compared with 50% for optimised referral and 99% for testing all breast cancers (Figure 2). The proportion of unaffected female relatives with a P/LP variant who were identified through genetic testing was estimated to be 42% for current practice, 55% for optimised referral, and 85% for testing all breast cancers. For all scenarios, the majority (70–77%) of P/LP variants in relatives were found through predictive testing. The remaining 23–30% of carriers were only identified after they developed cancer in the years following the proband’s diagnosis. *PALB2* carriers had the greatest increase in P/LP variant detection rates through expanded testing, starting at 21% for current practice, 34% for optimised referral, and 80% for testing all breast cancers. In contrast, *BRCA1* P/LP variant detection increased from 74% to 91%, and *BRCA2* detection increased from 45% to 87% for current practice and testing of all breast cancers, respectively.

#### 3.2.3. Cancer Incidence and Mortality

Increasing genetic referral and expanded testing resulted in higher estimated rates of risk-reducing surgery (Figure 3) and lower incidence of breast and ovarian cancer in relatives of probands with a P/LP variant in *BRCA1*, *BRCA2* or *PALB2* (Figure 4). 

The reduced cancer incidence translated to an improvement in overall estimated life expectancy and a reduction in the number of cancer-specific deaths (Figure 5). There was no difference in breast or ovarian survival between scenarios for relatives diagnosed with cancer (Appendix A). For probands with a P/LP variant and their unaffected female relatives, the average life expectancy increased by 0.122 years from optimised referral of all breast and ovarian cancer cases compared to current practice. Life-years saved were estimated from the timepoint of the proband’s breast cancer diagnosis (the start of the genetic testing intervention). For the same group, additional life-years saved through optimised referral were 0.127 compared to current practice. When comparing testing all breast cancer to optimised referral, although there was an increase in average life expectancy (0.126 years), there was a decrease in life-years saved of 0.231. A similar pattern was seen when limited to only relatives with a P/LP variant (Table 4). 

#### 3.2.4. Sensitivity Analysis Outcomes

NEEMO estimated similar numbers to a previous study for eligible at-risk female relatives, with an average of 1.52–1.55 per proband in the current study compared to 1.41–1.46 relatives per proband [22]. The estimated additional increase in life expectancy of 374 days (undiscounted) for P/LP variant carriers was also similar to the previous study (289–419 days). Removing background diagnostic genetic testing in relatives combined with assuming 100% uptake of predictive testing led to scenario 4 (genetic testing of all breast cancers), generating a gain in life-years saved for both groups (Appendix A).

The underlying prevalence of P/LP variants in probands with breast cancer remained constant across all scenarios at 3.49–3.56%. There was a substantial difference in P/LP variant detection rates when compared with the original NEEMO detection rates shown in Figure 2. For example, while only 13.9% of *BRCA1* P/LP variants remained undetected for the original scenario 3 (optimised referral), this increased to 55.6% in the sensitivity analysis (Figure 6). Similarly, for the original scenario 4 (testing all breast cancers), 14% of P/LP variants in the relatives were never detected, compared to only 1% in the sensitivity analysis. The consequence of these changes was a much greater incremental difference in detection rates for relatives between scenarios 3 and 4 in the sensitivity analysis.

## 4. Discussion

NEEMO is an innovative and adaptable microsimulation model for performing genetic testing evaluations in hereditary breast and ovarian cancer. The model incorporates modifiable predictive genetic testing elements and family dynamics, as well as gene-specific breast/ovarian cancer natural histories and cancer risk management adherence. A main strength of the model is that it is more likely to reflect current Australian practice than previously reported models, and it captures a broader group of individuals: female probands, first- and second-degree relatives, including male family members. Several outputs were externally validated, including mortality, cancer pathology, predictive testing uptake, and genetic testing referral rates. It is the most detailed model to date for modelling familial cancer clustering and family member interactions within an HBOC setting. 

While the current evaluation reports estimates for the Australian population and health services, NEEMO can be adapted to other settings. Many of the model inputs, such as those related to gene-specific pathology, cancer incidence, and survival, are likely to be generalisable to other settings as currently specified, as, by necessity, these inputs were largely informed by international cohort data. Other country-specific estimates, such as background cancer incidence, estimated gene prevalence and family size parameters, can be modified to reflect non-Australian populations if appropriate data are available. Selected modelled assumptions around clinical pathways, access to risk mitigation strategies, and uptake of genetic testing would also need to be modified, for example, introducing a delay for genetic testing following a cancer diagnosis instead of assuming same-year testing. This applies particularly in low-resource settings, where access to genetic testing may be more limited [46].

Adherence to cancer risk management has previously been shown to influence clinical and cost-effectiveness outcomes [27,28,47]. For the current analysis, the estimated uptake rates for risk-reducing surgery in unaffected relatives who tested positive for a P/LP variant were comparable to those seen in both genetics services and P/LP variant carriers ascertained through a population-based cohort (Appendix A). As a limitation, while the model allows for increased uptake of CRRM in women diagnosed with breast cancer who are identified as carrying a P/LP variant, it does not currently include other potential treatment effects such as improved progression-free survival via PARP-inhibitors [48,49] or improved survival from higher rates of systemic treatment [50].

NEEMO was applied to four alternative genetic testing scenarios, with clinical outcomes presented for probands who were diagnosed with breast cancer during the simulation and their first- and second-degree relatives. Our estimates indicate an increased life expectancy and reduced cancer burden for current practice compared to no genetic testing, and further improvement when all breast and ovarian cancer cases that would be eligible for genetic testing are referred to testing services. Expanding testing to all breast cancer cases further reduced breast and ovarian cancer incidence, but this did not translate to an improvement in life-years saved compared to optimised referral alone.

Our estimates can be compared with a previous study evaluating genetic testing for all breast cancer cases by Sun et al. [22], which did not directly model relatives and predictive testing, but instead used calculations based on average numbers of relatives and their expected age distributions. Due to the difference between the models, assumptions were introduced into NEEMO to assist comparison by removing background diagnostic genetic testing in relatives and maximising the uptake of predictive testing. With these assumptions in place, both models estimated similar numbers of eligible at-risk female relatives and estimated increases in life expectancy for expanding testing to all breast cancers.

Unlike previous models, NEEMO accounts for a background increase in genetic testing, as would be expected under different testing scenarios. For each scenario, probands with a P/LP variant detected prior to their cancer diagnosis were excluded from the outcomes presented in Table 3 and Table 4. The impact of background genetic testing was evident in the decreasing prevalence of P/LP variants in the newly diagnosed probands, corresponding to the increase in genetic testing across different scenarios (3.53% for no testing and 2.39% for testing all breast cancers). An earlier study reported that treatment-focused testing of all incident cancer cases could theoretically identify most P/LP variant carriers in the US population within 14 years if at least half of the relatives underwent predictive testing [51]. A clear flow-on effect of this is that the P/LP variant detection rate in incident cancers would decrease over time. These diminishing returns for expanding genetic testing criteria are effectively captured within NEEMO. 

### Limitations

There are several limitations to the model. When modelling current practice, the selection of women eligible for genetic testing is simplified (based on subtypes, age, and family history). In practice, quantitative tools such as CanRisk utilise much more detailed personal and family information to estimate carrier probabilities tied to eligibility for genetic testing [52]. An advantage of microsimulation models such as NEEMO is that aspects such as these can be further refined over time, bringing in more personalised data and improving the precision of model estimates [53].

Variants of uncertain significance (VUS) and potential for reclassification have not yet been included. A study on mainstream genetic testing of breast cancer patients in Australia identified VUS in 8.7% of breast cancer patients who underwent genetic testing [54]. The addition of VUS would more accurately reflect clinical practice; therefore, this is planned to be incorporated in future iterations of NEEMO.

Limits were placed on family size as described in the methods, and variation in fertility and cancer incidence over calendar time is not included. In practice, genetic testing may occur many years after the incident cancer diagnosis, but the model assumes that any P/LP variant detection would occur in the same year as the diagnosis. The shift towards treatment-focused testing for breast and ovarian cancer will potentially make this less of an issue in the future [45].

## 5. Conclusions

NEEMO is an adaptable and validated microsimulation model for evaluating genetic testing interventions for hereditary breast and ovarian cancer. It allows for varying uptake of predictive genetic testing over time and cancer risk management adherence, which are important considerations when considering real-world clinical and cost-effectiveness, as well as guiding potential future implementation. Further applications of the model include the evaluation of the cost-effectiveness of alternative genetic testing protocols, including population-based testing for both high-risk genes, as well as the expanded gene panel. 

## Figures and Tables

**Figure 1 cancers-16-04165-f001:**
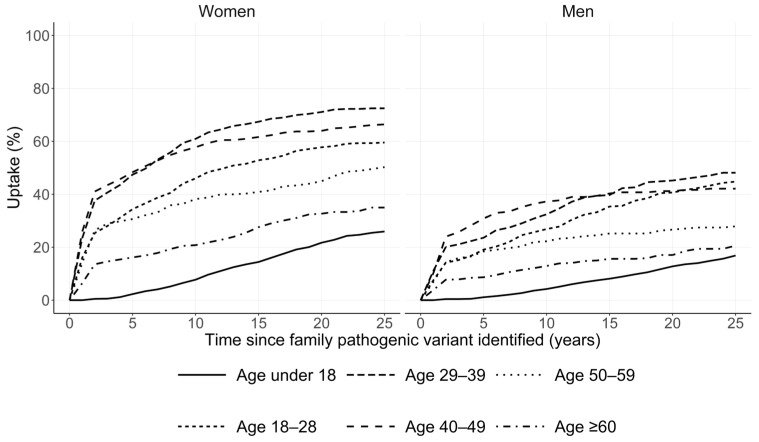
Simulated uptake of predictive testing in relatives. Uptake of predictive genetic testing is shown for all first- and second-degree relatives by age and sex. Age groups are based on the relatives’ age at the time the family P/LP variant was first identified. Individuals were censored at age at death, age at full P/LP variant detection (clinically indicated genetic testing using a full gene panel), or age at which they exited the model. Abbreviations: P/LP, pathogenic/likely pathogenic.

**Figure 2 cancers-16-04165-f002:**
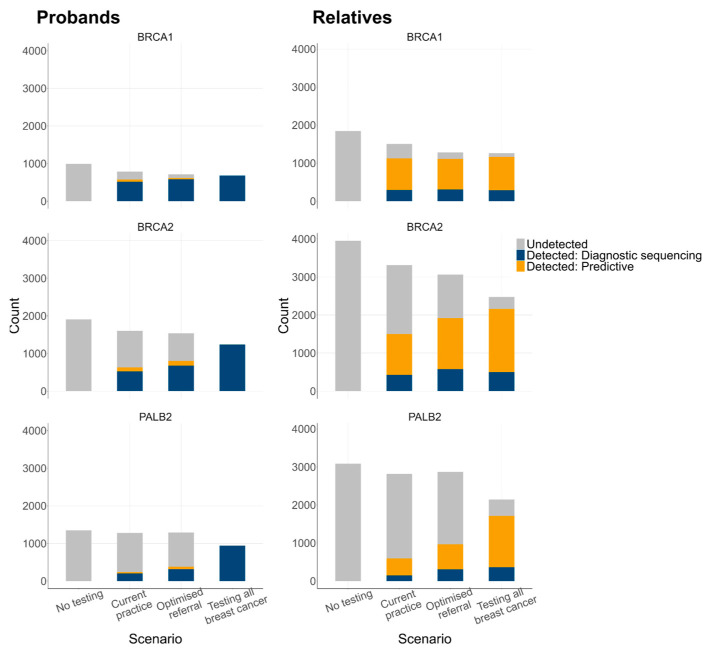
P/LP variant detection by scenario and gene. Relatives were female relatives with a P/LP variant, unaffected by cancer at the time of the proband’s breast cancer diagnosis.

**Figure 3 cancers-16-04165-f003:**
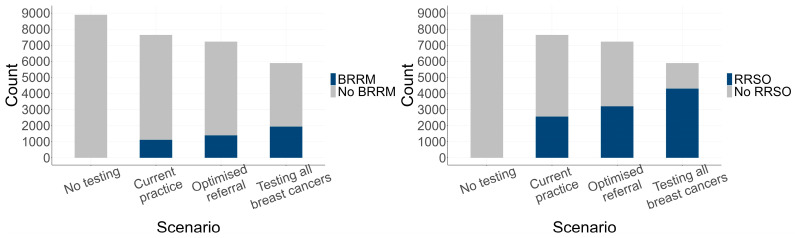
Risk-reducing surgery uptake in relatives with a P/LP variant. Relatives were female relatives with a P/LP variant, unaffected by cancer at the time of the proband’s breast cancer diagnosis. Abbreviations: BRRM, bilateral risk-reducing mastectomy; P/LP, pathogenic/likely pathogenic; RRSO, risk-reducing salpingo-oophorectomy.

**Figure 4 cancers-16-04165-f004:**
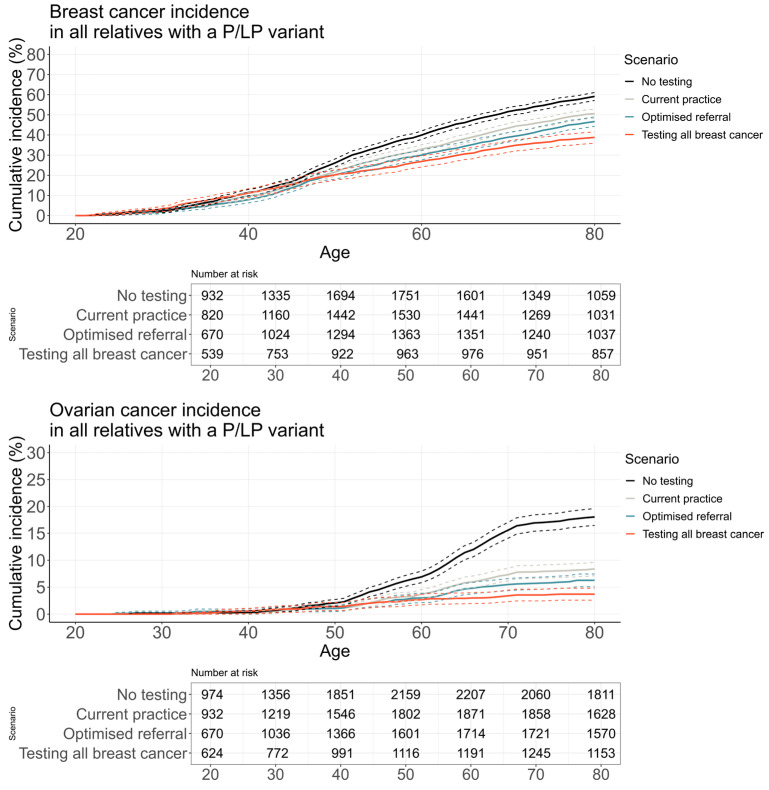
Estimated cancer incidence among female relatives with a P/LP variant. Cancer incidence is shown among first- and second-degree female relatives of a breast cancer-affected proband with a P/LP variant in *BRCA1*, *BRCA2* or *PALB2*, who were unaffected by breast/ovarian cancer at entry. Entry time was the year of the proband’s first breast cancer diagnosis. Women were censored at the age of breast cancer, ovarian cancer, other-cause death, or the age at which they exited the model (maximum age 100). Abbreviations: P/LP, pathogenic/likely pathogenic.

**Figure 5 cancers-16-04165-f005:**
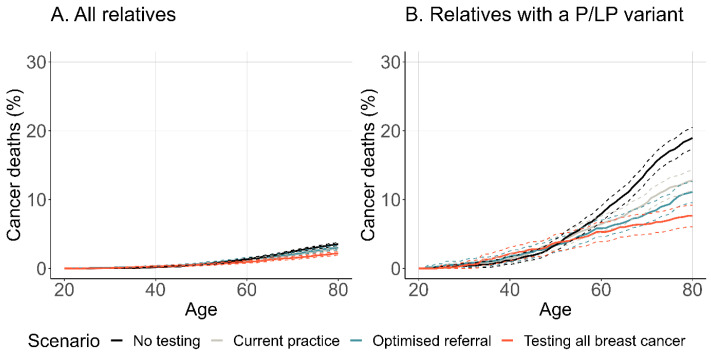
Estimated death from breast or ovarian cancer in female relatives. Female relatives were included if they were related to a breast cancer-affected proband with a P/LP variant in *BRCA1*, *BRCA2* or *PALB2*, and unaffected by cancer at entry. Entry time was the year of the proband’s initial breast cancer diagnosis. Events were breast or ovarian cancer-specific death. Women were censored at other-cause death or the age at which they exited the model (maximum age 100). (**A**) includes all first- and second-degree unaffected female relatives of P/LP carriers. (**B**) includes only unaffected female relatives who carry a P/LP variant in *BRCA1*, *BRCA2* or *PALB2*. Abbreviations: P/LP, pathogenic/likely pathogenic.

**Figure 6 cancers-16-04165-f006:**
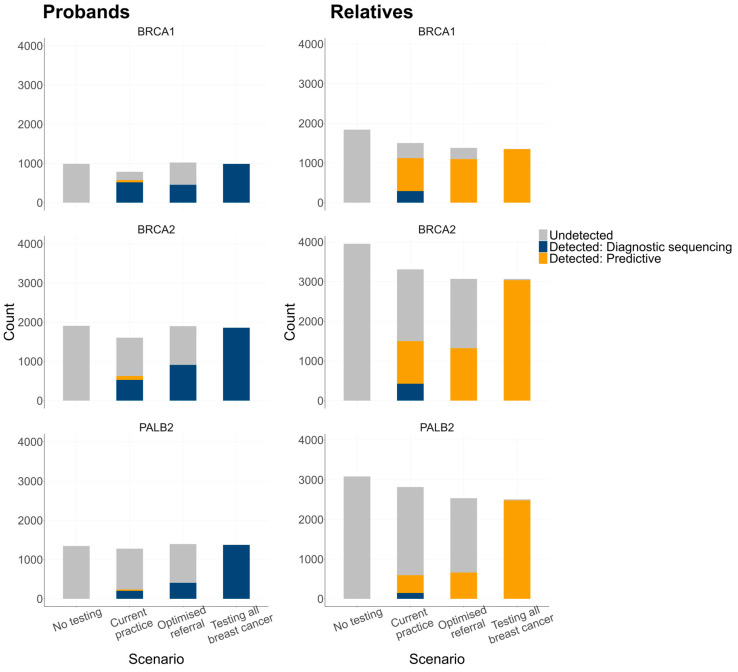
P/LP variant detection by scenario in the sensitivity analysis. Diagnostic genetic testing in relatives was removed for both the optimised referral and testing of all breast cancer scenarios for sensitivity analysis. The model also assumed a 100% uptake of predictive testing in relatives of probands that were found to carry a P/LP variant. The no-testing and current practice scenarios are shown for comparison but were not altered for the sensitivity analysis. Relatives were female relatives with a P/LP variant, unaffected by cancer at the time of the proband’s breast cancer diagnosis. Abbreviations: P/LP, pathogenic/likely pathogenic.

**Table 1 cancers-16-04165-t001:** Genetic testing referral rates and uptake of genetic testing.

		Odds Ratio	Source
Breast cancer referral			
	Constant	3.23	Parkville Familial Cancer Centre, the Victorian Cancer Registry, expert opinion
	Aged 40–49	0.38	
	Aged 50–69	0.22	
	Aged 70 and over	0.10	
	Survival < 12 months	0.35	
	High grade	2.37	
	Non triple-negative	0.50	Assumption (expert opinion)
	HER2	0.22	Assumption (expert opinion)
Ovarian cancer referral			
	Constant	3.09	Parkville Familial Cancer Centre, the Victorian Cancer Registry, expert opinion
	Aged 60–74	0.20	
	Aged 75 and over	0.08	
	Survival < 12 months	0.04	
	High grade	1.06	
	Prior breast cancer diagnosis ^1^	17.75	
	Mucinous histotype	0.08	Assumption (expert opinion)
	Other histotype ^2^	0.20	Assumption (expert opinion)
Uptake			
	Breast cancer	0.96	[10]
	Ovarian cancer	0.95	[11]

^1^ All women with prior breast cancer diagnoses (*n* = 58) within the sample were referred to genetics services.^2^ Refers to ovarian cancer that is neither serous, clear cell, endometrioid, or mucinous.

**Table 2 cancers-16-04165-t002:** Predictive genetic testing probabilities.

		Time Since Family Pathogenic/Likely Pathogenic Variant First Identified			Source
**First Degree**		**<1 Year**	**1–3 Years**	**>3 Years**	Parkville Familial Cancer Centre, unpublished observation
	Age < 18	0	0	0	
	Age 18–29	0.259	0.098	0.071	
	Age 30–49	0.286	0.098	0.022	
	Age 50–59	0.146	0.033	0.010	
	Age ≥ 60	0.099	0.014	0.007	
**Second Degree**					
	Age <18	0	0	0	
	Age 18–29	0.072	0.045	0.034	
	Age 30–49	0.112	0.080	0.021	
	Age 50–59	0.114	0.034	0.002	
	Age ≥ 60	0.039	0.017	0.002	

A rate ratio of 0.422 was applied to male relatives’ uptake of predictive genetic testing.

**Table 3 cancers-16-04165-t003:** Genetic outcomes in probands diagnosed with breast cancer and their relatives.

Group	Outcome		Scenario 1: No Genetic Testing	Scenario 2: Current Practice	Scenario 3: Optimised Referral of Breast and Ovarian Cancer	Scenario 4: Genetic Testing for all Breast Cancers
Probands (women diagnosed with breast cancer before the age of 80 years)	Total number		121,343	120,700	121,299	120,166
	Age at diagnosis, mean (sd)		60.53 (10.93)	60.5 (10.94)	60.47 (10.92)	60.60 (10.90)
	Prevalence of P/LP variants, n (%)	*BRCA1* *BRCA2* *PALB2* Total	990 (0.82%) 1908 (1.58%) 1347 (1.11%) 4245 (3.53%)	787 (0.65%) 1605 (1.33%) 1278 (1.05%) 3670 (3.05%)	714 (0.59%) 1539 (1.28%) 1291 (1.06%) 3544 (2.95%)	684 (0.56%) 1235 (1.02%) 948 (0.78%) 2867 (2.39%)
	Genetic testing, diagnostic sequencing, n (%)		0 (0%)	9279 (7.69%)	16,159 (13.32%)	120,162 (99.99%)
	Genetic testing, predictive test, n (%)		0 (0%)	537 (0.44%)	727 (0.60%)	664 (0.55%)
	Secondary ovarian cancer in P/LP variant carriers, n (%)		557 (13.12%)	283 (7.71%)	200 (5.64%)	49 (1.71%)
Male and female relatives of probands who carry a P/LP variant	Total number		66 358	56,672	54,020	43,352
	Average per proband		15.63	15.44	15.24	15.12
	Genetic testing, diagnostic sequencing, n (%) ^1^		0 (%)	1438 (2.54%)	2238 (4.14%)	3938 (9.08%)
	Genetic testing, predictive test, n (%) ^1^		0 (0%)	10,813 (19.09%)	13,705 (25.37%)	19,621 (45.26%)
All cancer-unaffected female relatives, including non-carriers	Total number		30,736	26,661	25,319	20,461
	Average per proband		7.24	7.26	7.14	7.14
	Age, median (IQR)		34 (18, 58)	34 (18, 59)	34 (17, 59)	34 (17, 60)
Cancer-unaffected female relatives, excluding non-carriers	Total number		8881	7633	7217	5881
	Average per proband		2.08	2.07	2.06	2.04
	Age, median (IQR)		31 (17, 48)	31 (17, 49)	31 (17, 50)	31 (17, 51)

^1^ These include diagnostic sequencing genetic tests and predictive tests unrelated to proband genetic testing or proband breast cancer diagnosis (e.g., another non-proband relative is found to carry a P/LP variant). For example, of the 1438 diagnostic sequencing detections under the current practice scenario, 1181 (82%) were women unaffected at the time of the proband breast cancer diagnosis but who were later diagnosed with cancer, with the remainder being women tested due to a personal cancer diagnosis prior to the proband’s diagnosis. Abbreviations: IQR, interquartile range; P/LP, pathogenic/likely pathogenic.

**Table 4 cancers-16-04165-t004:** Life expectancy and life-years saved by genetic testing scenario.

Group	Outcome	Scenario 1: No Genetic Testing		Scenario 2: Current Practice		Scenario 3: Optimised Referral of Breast and Ovarian Cancer		Scenario 4: Genetic Testing all Breast Cancers	
		Mean	95% CI	Mean	95% CI	Mean	95% CI	Mean	95% CI
Probands with a P/LP and their relatives ^1^	Life expectancy	84.672	(84.538, 84.806)	85.067	(84.927, 85.208)	85.189	(85.045, 85.333)	85.315	(85.155, 85.475)
	Life-years saved	41.903	(41.606, 42.199)	41.991	(41.671, 42.311)	42.118	(41.787, 42.449)	41.887	(41.519, 42.255)
Relatives ^1^ with a P/LP variant only	Life expectancy	82.304	(82.008, 82.600)	83.443	(83.141, 83.746)	83.973	(83.667, 84.279)	84.159	(83.818, 84.501)
	Life-years saved	47.860	(47.373, 48.347)	48.780	(48.241, 49.319)	49.150	(48.595, 49.705)	49.112	(48.495, 49.730)

^1^ Included relatives were female, and unaffected by cancer at the time of the proband’s breast cancer diagnosis. Abbreviations: CI, confidence interval; P/LP, pathogenic/likely pathogenic.

## Data Availability

The data used in this study for breast and ovarian cancer model parameters can be found at: https://seer.cancer.gov (accessed 29 January 2020). Other data that were used to support the development of the microsimulation model are not openly available due to reasons of sensitivity and are available from the corresponding author upon reasonable request, contingent on approval by a Human Research Ethics Committee. Data are located in controlled access storage at the Peter MacCallum Cancer Centre. The code for the microsimulation model is a proprietary property and cannot be provided directly by the authors. For access and potential collaboration, interested researchers should contact the corresponding author in the first instance.

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
