# Peer review of "Reduced Breast and Ovarian Cancer Through Targeted Genetic Testing: Estimates Using the NEEMO Microsimulation Model"

_cancers, 2024, doi:10.3390/cancers16244165_

Round 1
Reviewer 1 Report
Comments and Suggestions for Authors
The manuscript describes the development of NEEMO-a microsimulation model intended for the evaluation of HBOC genetic testing strategies, in particular the cancer predisposition genes BRCA1 and BRCA2, and PALB2. It simulates family structures and cancer histories and integrates genetic, demographic, and clinical data in predicting a number of relevant outcomes for four different genetic testing strategies. The results imply that an extended genetic testing of all cases with breast cancer can be associated with a decrease in cancer incidence and increased life expectancy in unaffected relatives. Compared to previous models, NEEMO is novel in that it models family dynamics and predictive testing uptake, providing a more finely tuned tool for the evaluation of the clinical and cost-effectiveness of different testing strategies for HBOC.
Major comments:
Limitations of the Model: While NEEMO did capture detailed genetic and familial inputs, it didn't really allow for considerations of variants such as VUS, nor did it account for the potential of reclassification. Addition of these could increase model precision since VUS are common in the real world of genetic testing.
Simplified Eligibility and Testing Assumptions: Assumptions from current models are not reflective of real clinical timelines and long-term predictions, such as the same-year genetic testing following diagnosis. Expanding inclusions to cover delayed testing timelines and incorporating tools such as CanRisk for personalized probabilities could afford refinement in eligibility assessment.
Current Treatment Strategies for BRCA-mutated Breast Cancer Patients: Lines 78-80. The sentence "P/LP variant status in women already affected by cancer can influence treatment decisions, such as opting for mastectomy over breast conserving surgery, and increasing access to targeted or systemic therapy." is not completely true. In fact, many surgical treatment strategies exist for BRCA-mutated breast cancer patients, each with its own prognostic implications. Please cite PMID: 35534308 to give more clinical context to your study and improve your manuscript, overall.
Future Applications and Generalizability: This manuscript might also consider how NEEMO might be fitted for non-Australian settings, detailing what country-specific adjustments might be necessary.
Minor comments:
Introduction section is too long. Reduce.
Author Response
Comments 1: Limitations of the Model: While NEEMO did capture detailed genetic and familial inputs, it didn't really allow for considerations of variants such as VUS, nor did it account for the potential of reclassification. Addition of these could increase model precision since VUS are common in the real world of genetic testing.
Response 1: We agree that variants of uncertain significance (VUS) are an important consideration when modelling genetic testing interventions, particularly at a population level. They are not recommended to be used to determine clinical care. A study of mainstream genetic testing of breast cancer patients in Australia identified VUS in 8.7% of breast cancer patients who underwent genetic testing (Beard et al. PMID 33723355). This estimate is likely to change in future with an increase in the number of genes being tested for (including moderate risk genes), increased testing in underrepresented ethnicities, and due to initiatives such as ClinGen’s Variant Curation Expert Panels. Due to this complexity, we excluded consideration of VUS from the current study. Mersch et al. found only about 9% of reclassified unique VUS were upgraded to pathogenic or likely pathogenic, affected approximately 0.1% of individuals tested (PMID 30264118). In future iterations of NEEMO we intend to do further sensitivity analyses, which will include aspects such as VUS rates and the impact on health and cost outcomes.
We have expanded on the limitations described in the discussion, added the following test to page 17, line 580 of the manuscript:
“A study of mainstream genetic testing of breast cancer patients in Australia identified VUS in 8.7% of breast cancer patients who underwent genetic testing (Beard et al. 2021). The addition of VUS would more accurately reflect clinical practice so will be incorporated in future iterations of NEEMO.”
Comments 2: Simplified Eligibility and Testing Assumptions: Assumptions from current models are not reflective of real clinical timelines and long-term predictions, such as the same-year genetic testing following diagnosis. Expanding inclusions to cover delayed testing timelines and incorporating tools such as CanRisk for personalized probabilities could afford refinement in eligibility assessment.
Response 2:
Part A) CanRisk:
Thank you, we agree that tools like CanRisk would add value to the model and improve precision and predictions. We have added the following text to the limitations section at page 16 line 574 of the manuscript to expand on this. It now reads:
“In practice, quantitative tools such as CanRisk utilise much more detailed personal and family information to estimate carrier probabilities tied to eligibility for genetic test-ing.52 An advantage of microsimulation models such as NEEMO is that aspects such as these can be further refined over time, bringing in more personalised data and improving the precision of modelled estimates (Kopaskar et al 2023).”
It is also worth noting that as HBOC genetic testing is increasingly ordered by oncologists through a mainstreaming model utilisation of specialised tools such as CanRisk may be more difficult in the real clinical setting. Patients often do not know their family cancer histories, or clinicians are unable to verify significant cancer histopathologies. As such, not enough details may be collected for a reliable estimate of carrier probability.
Part B) Clinical testing timelines:
Regarding the assumption of same-year genetic testing, the reviewer raises a valid point that it may not reflect real clinical timelines. This is especially true in low-resource settings where access to genetic testing is less readily available directly after diagnosis. Within the Australian setting, mainstream genetic testing is being more routinely offered because of the impact on access to targeted treatment such as PARP-inhibitors for both breast and ovarian cancer (PMID 34081848, PMID 30345884). We could find limited published data on time to genetic testing from initial cancer diagnosis in Australia. Crispin et al (PMID: 30251447) published the time to genetic testing results from referral for breast cancer patients aged <40 years at diagnosis. This went from median of 24 months (IQR 4-132) in the years 2001-2005, to a median of 5 months (IQR 3-10) during the years 2011-2016. Therefore, for the more recent time period at least 75% of women had genetic testing within a year of their diagnosis.
We consider this aspect might have increased importance if NEEMO is to be adapted to a setting with more restricted access to genetic testing. To assess this issue we updated the model to allow for a delay in genetic testing of up to 10 years from a patient’s breast or ovarian cancer diagnosis. We ran the model with this update looking at the current practice scenario only, as it is still assumed best practice under the optimised referral and testing all breast cancer scenarios would be to undergo genetic testing close to the time of diagnosis. We used unpublished observed data from the Parkville Familial Cancer Centre for the time to genetic testing for BRCA1/2 carriers diagnosed with cancer from 2010 onwards, and assumed all genetic testing would be within 10 years following diagnosis.
The outcomes of the model are shown in the table below. Briefly, incorporating up to a 10-year delay in genetic testing did not significantly alter the outcomes reported in the original Table 3 (now split into Tables 3 and 4). Similarly, uptake of risk-reducing surgery and cancer outcomes in unaffected relatives were not significantly different. Estimates for life expectancy were within the 95% confidence intervals of the original values in Table 4 (now Table 5: Clinical outcomes in relatives by genetic testing scenario). The only value to show variation was the estimate for life years saved for all women (i.e. including all probands and all relatives irrespective of carrier status), where the estimate was slightly lower at 41.844 life years saved (95% CI 41.797, 41.891) compared to the original 42.001 (95% CI 41.955, 42.048). Given this would have no impact on the conclusions drawn from the study we suggest not including these results in the current manuscript (particularly due to the lack of robust input data), but to instead add the following comment related to future applications and generalisability at page 16, line 528:
“Selected modelled assumptions around clinical pathways and uptake of genetic testing would also need to be modified, for example introducing a delay for genetic testing following a cancer diagnosis instead of assuming same-year testing. This applies particularly in low-resource settings where access to genetic testing may be more limited (Manchanda et al 2020).”
Table 3. Genetic outcomes in women diagnosed with breast cancer.
|
Probands Women diagnosed with breast cancer before age 80 |
|
Current practice (incorporating genetic testing delay)
|
Current practice (original)
|
||
|
|
|
N |
% |
N |
% |
|
Total number |
|
121 325 |
100 |
120 700 |
100 |
|
Number of P/LP variant carriers |
BRCA1 |
807 |
0.67 |
787 |
0.65 |
|
|
BRCA2 |
1547 |
1.28 |
1605 |
1.33 |
|
|
PALB2 |
1316 |
1.50 |
1278 |
1.05 |
|
|
Total |
3670 |
3.13 |
3670 |
3.05 |
|
Age at breast cancer diagnosis, mean (sd) |
|
60.6 (10.92) |
|
60.5 (10.94) |
|
|
Genetic testing, full sequencing |
|
9963 |
8.21 |
9279 |
7.69 |
|
Genetic testing, predictive tests |
|
628 |
0.52 |
537 |
0.44 |
|
P/LP variant detected |
Full sequencing |
1169 |
31.85 |
1234 |
33.62 |
|
|
Predictive test |
214 |
5.83 |
188 |
5.12 |
|
P/LP variant carriers with secondary ovarian cancer |
|
324 |
8.83 |
283 |
7.71 |
Table 4. Genetic outcomes in the relatives of probands who carry a P/LP variant
|
First- and second-degree relatives of probands who have a P/LP variant |
|
Current practice (incorporating genetic testing delay)
|
Current practice (original) |
||
|
|
|
N |
% |
N |
% |
|
Total female Total male and female |
|
28 507 55 270 |
100 100 |
29 467 56 672 |
100 100 |
|
Number of full sequencing genetic tests |
|
1528 |
5.36 |
1438 |
4.88 |
|
Number of predictive tests (male and female relatives) |
|
11966 |
21.65 |
10 813 |
19.08 |
|
Number of female relatives unaffected by cancer at the time of proband diagnosis
Averages are per proband with a P/LP variant |
|
N |
Average |
N |
Average |
|
Total |
25 861 |
7.04 |
26 661 |
7.26 |
|
|
Median age (IQR) |
34 (18, 59) |
|
34 (18, 59) |
|
|
|
Number of female relatives with a P/LP variant, unaffected by cancer at the time of proband diagnosis |
|
N |
Average |
N |
Average |
|
Total |
7435 |
2.03 |
7633 |
2.07 |
|
|
Median age (IQR) |
31 (17, 50) |
|
31 (17, 49) |
|
|
|
|
|
N |
% |
N |
% |
|
Distribution of genotype in unaffected female relatives with a P/LP variant |
BRCA1 |
1472 |
19.80 |
1504 |
19.70 |
|
BRCA2 |
3050 |
41.02 |
3310 |
43.36 |
|
|
PALB2 |
2913 |
39.18 |
2819 |
36.93 |
|
|
Total |
7435 |
100 |
7633 |
100 |
|
|
P/LP variant detected during lifetime |
BRCA1 |
1134 |
77.04 |
1109 |
73.74 |
|
BRCA2 |
1447 |
47.44 |
1477 |
44.62 |
|
|
PALB2 |
603 |
20.70 |
594 |
21.07 |
|
|
Total |
3184 |
42.82 |
3180 |
41.66 |
|
|
|
|
Full sequence |
Predictive test |
Full sequence |
Predictive test |
|
Method of gene detection (%) |
All ages |
27.67 |
72.33 |
26.73 |
73.27 |
Table 5. Clinical outcomes in relatives by genetic testing scenario.
|
Outcome |
Current practice (incorporating genetic testing delay)
|
Current practice (original) |
|||
|
|
|
N |
% |
N |
% |
|
Risk-reducing surgery and cancer incidence in unaffected relatives
|
Total at risk |
7435 |
100 |
7633 |
100 |
|
Bilateral risk-reducing mastectomy |
1057 |
14.22 |
1111 |
14.56 |
|
|
Risk-reducing salpingo-oophorectomy |
2513 |
33.80 |
2560 |
33.54 |
|
|
Breast cancer diagnoses |
3216 |
43.25 |
3412 |
44.70 |
|
|
Ovarian cancer diagnoses |
673 |
9.08 |
684 |
8.96 |
|
|
|
|
Mean |
95% CI |
Mean |
95% CI |
|
Life expectancy |
All women |
86.565 |
(86.545, 86.585) |
86.548 |
(86.528, 86.568) |
|
All probands with a P/LP variant, and their unaffected relatives |
85.060 |
(84.918, 85.202) |
85.067 |
(84.927, 85.208) |
|
|
Only relatives of probands with a P/LP variant |
85.752 |
(85.611, 85.893) |
85.810 |
(85.672, 85.949) |
|
|
Relatives with a P/LP variant |
83.392 |
(83.083, 83.701) |
83.443 |
(83.141, 83.746) |
|
|
Life years saved |
All women |
41.844 |
(41.797, 41.891) |
42.001 |
(41.955, 42.048) |
|
All probands with a P/LP variant, and their unaffected relatives |
41.648 |
(41.323, 41.973) |
41.991 |
(41.671, 42.311) |
|
|
Only relatives of probands with a P/LP variant |
47.558 |
(47.248, 47.868) |
47.771 |
(47.466, 48.075) |
|
|
Relatives with a P/LP variant |
48.550 |
(48.006, 49.094) |
48.780 |
(48.241, 49.319) |
|
Comments 3: Current Treatment Strategies for BRCA-mutated Breast Cancer Patients: Lines 78-80. The sentence "P/LP variant status in women already affected by cancer can influence treatment decisions, such as opting for mastectomy over breast conserving surgery, and increasing access to targeted or systemic therapy." is not completely true. In fact, many surgical treatment strategies exist for BRCA-mutated breast cancer patients, each with its own prognostic implications. Please cite PMID: to give more clinical context to your study and improve your manuscript, overall.
Response 3: Thank you, we have added the suggested reference to page 2, lines 78-80 to provide more context. We have added an additional reference for the Australian context, where P/LP variant status led to changes in recommended breast cancer management for 24/31(77%) women with recent breast cancer diagnoses (PMID 37005005).
Comments 4: Future Applications and Generalizability: This manuscript might also consider how NEEMO might be fitted for non-Australian settings, detailing what country-specific adjustments might be necessary.
Response 4: Thank you. We have expanded on the comments on generalisability in the discussion at page 15, lines 523-532, including the addition regarding the assumption of genetic testing the same year as cancer diagnosis as detailed above. This section now reads:
“While the current evaluation reports estimates for the Australian population and health services, NEEMO can be adapted to other settings. Many of the model inputs, such as those related to gene-specific pathology, cancer incidence and survival are likely to be generalisable to other settings as currently specified, as by necessity these inputs were largely informed from international cohort data. Other country-specific estimates, such as background cancer incidence, estimated gene prevalence and family size parameters can be modified to reflect non-Australian populations if accurate data are available. Selected modelled assumptions around clinical pathways, access to risk mitigation strategies, and uptake of genetic testing would also need to be modified, for example introducing a delay for genetic testing following a cancer diagnosis instead of assuming same-year testing. This applies particularly in low-resource settings where access to genetic testing may be more limited (Manchanda et al 2020).”
Comments 5: Introduction section is too long. Reduce.
Response 5: We have reduced the introduction where possible by 53 words, but are concerned further reduction would result in insufficient background and context for the study, particularly as Reviewer 2 felt this aspect was adequately addressed.
Reviewer 2 Report
Comments and Suggestions for Authors
Manuscript ID: cancers-3288757
Title: Reduced breast and ovarian cancer through targeted genetic testing: estimates using the NEEMO microsimulation model
Authors: Lara Petelin *, Michelle Cunich, Pietro Procopio, Deborah Schofield, Lisa Devereux, Carolyn Nickson, Paul A James, Ian G Campbell, Alison H Trainer Cancer Epidemiology and Prevention
This study introduces the NEEMO microsimulation model, designed to assess the clinical and economic impact of genetic testing strategies for hereditary breast and ovarian cancer in Australia. NEEMO simulates genetic testing and cancer risk management within five-generation family structures, incorporating both monogenic and polygenic risks. The model evaluates four genetic testing strategies: no testing, current practice, optimized referral, and universal testing for breast cancer (BC) patients. Results indicate that optimized genetic testing and expanded referral prevent more BC and ovarian cancers among relatives compared to no testing or universal testing for BC patients, without significant improvement in life years saved beyond optimized referral.
The main critiques focus on the presentation of the results, particularly in the tables. The tables contain too much information, making them difficult to interpret. For instance, on page 12 in Table 3, the row for “First and second-degree relatives of probands who have a P/LP variant” is entirely empty, while the following two rows display numbers and ratios. It’s unclear how these ratios were calculated, especially in relation to the provided numbers.
Another point of concern is the item, "the Number of female relatives of proband P/LP variant carriers, unaffected by cancer at the time of proband diagnosis. Averages are per proband with a P/LP variant." I would like clarification on the number of female relatives of proband P/LP variant carriers who were unaffected by cancer at the time of the proband’s diagnosis, as well as those who were affected. Additionally, in terms of the "Averages per proband with a P/LP variant," it’s unclear how these values were calculated. Similarly, for the item below "Number of female relatives with a P/LP variant, unaffected by cancer at the time of proband diagnosis,"
For the item "P/LP variant detected during lifetime," could you clarify the method used to calculate the percentage? Specifically, I would like to understand the denominator used for this calculation.
The same concern applies to Table 4: it is unclear how the percentages were calculated. Could you clarify the denominator used for these calculations? This information is essential for accurately interpreting the rates presented in the table.
Regarding the items, it’s difficult to understand the distinctions between "All probands with a P/LP variant and their unaffected relatives," "Only relatives of probands with a P/LP variant," and "Relatives with a P/LP variant." Could you clarify the differences among these categories?
Author Response
Thank you very much for taking the time to review this manuscript. Please find the detailed responses below and the corresponding revisions in tracked changes in the re-submitted files.
Comments 1: The main critiques focus on the presentation of the results, particularly in the tables. The tables contain too much information, making them difficult to interpret. For instance, on page 12 in Table 3, the row for “First and second-degree relatives of probands who have a P/LP variant” is entirely empty, while the following two rows display numbers and ratios. It’s unclear how these ratios were calculated, especially in relation to the provided numbers.
Response 1: We have split Table 3 into two tables (Table 3 and Table 4) to separate the outcomes for the probands from their relatives to make the presentation of these results clearer. We have simplified Table 4 by moving detail regarding the age-groups of the relatives to a table in the supplement (Supplemental Table S11). The total number of relatives (female only, and male+female) has been added to the new Table 4 to make the calculation of percentages clearer.
Comments 2: Another point of concern is the item, "the Number of female relatives of proband P/LP variant carriers, unaffected by cancer at the time of proband diagnosis. Averages are per proband with a P/LP variant."
I would like clarification on the number of female relatives of proband P/LP variant carriers who were unaffected by cancer at the time of the proband’s diagnosis, as well as those who were affected. Additionally, in terms of the "Averages per proband with a P/LP variant," it’s unclear how these values were calculated. Similarly, for the item below "Number of female relatives with a P/LP variant, unaffected by cancer at the time of proband diagnosis,"
Response 2: Thank you for the suggestions on how to improve this table. We have added two additional rows to the table (now Table 4: Genetic outcomes in the relatives of probands who carry a P/LP variant) that include both the total number of female relatives (unaffected and affected), as well as the total number of male and female relatives. We have also removed the age groups from this table so that it is less overwhelming. These are still available in the supplementary materials for readers who may be interested.
An additional footnote has been provided detailing the method of how the average per proband with a P/LP variant was calculated, and reads:
“3Averages are per proband with a P/LP variant. For example, for current practice there were 3670 proband P/LP variant carriers (see Table 3), and a total of 26 661 female relatives unaffected by cancer to give 7.26 relatives per proband on average.”
Table 4. Genetic outcomes in the relatives of probands who carry a P/LP variant
|
First- and second-degree relatives of probands who have a P/LP variant |
|
No genetic testing
|
Current practice
|
Optimised referral of breast and ovarian cancer |
Genetic testing for all breast cancers |
|||||
|
|
|
N |
% |
N |
% |
N |
% |
N |
% |
|
|
Number of full sequencing genetic tests 1 |
|
0 |
0 |
1438 |
4.88 |
2238 |
8.04 |
3938 |
17.71 |
|
|
Total number of female relatives2 |
|
34 169 |
100 |
29 482 |
100 |
27 815 |
100 |
22 237 |
100 |
|
|
Number of predictive tests (male and female relatives) |
|
0 |
0 |
10 813 |
19.08 |
13 705 |
25.37 |
19 621 |
45.26 |
|
|
Total number of male and female relatives |
|
66 358 |
100 |
56 672 |
100 |
54 020 |
100 |
43 352 |
100 |
|
|
|
|
N |
Average3 |
N |
Average3 |
N |
Average3 |
N |
Average3 |
|
|
Number of female relatives unaffected by cancer at the time of proband diagnosis |
Total |
30 736 |
7.24 |
26 661 |
7.26 |
25 319 |
7.14 |
20 461 |
7.14 |
|
|
Median age (IQR) |
34 (18, 58) |
|
34 (18, 59) |
|
34 (17, 59) |
|
34 (17, 60) |
|
|
|
|
Number of female relatives with a P/LP variant, unaffected by cancer at the time of proband diagnosis |
|
|
|
|
|
|
|
|
|
|
|
Total |
8881 |
2.08 |
7633 |
2.07 |
7217 |
2.06 |
5881 |
2.04 |
|
|
|
Median age (IQR) |
31 (17, 48) |
|
31 (17, 49) |
|
31 (17, 50) |
|
31 (17, 51) |
|
|
|
|
|
|
N |
% |
N |
% |
N |
% |
N |
% |
|
|
Distribution of genotype in unaffected female relatives with a P/LP variant |
BRCA1 |
1841 |
20.73 |
1504 |
19.70 |
1284 |
17.79 |
1263 |
21.48 |
|
|
BRCA2 |
3955 |
44.53 |
3310 |
43.36 |
3065 |
42.47 |
2477 |
42.12 |
|
|
|
PALB2 |
3085 |
34.74 |
2819 |
36.93 |
2868 |
39.74 |
2141 |
36.41 |
|
|
|
Total |
8881 |
100 |
7633 |
100 |
7217 |
100 |
5881 |
100 |
|
|
|
P/LP variant detected during lifetime4 |
BRCA1 |
0 |
0 |
1109 |
73.74 |
1103 |
85.9 |
1149 |
90.97 |
|
|
BRCA2 |
0 |
0 |
1477 |
44.62 |
1905 |
62.15 |
2147 |
86.68 |
|
|
|
PALB2 |
0 |
0 |
594 |
21.07 |
965 |
33.65 |
1710 |
79.87 |
|
|
|
Total |
0 |
0 |
3180 |
41.66 |
3973 |
55.05 |
5006 |
85.12 |
|
|
|
|
|
Full sequence |
Predictive test |
Full sequence |
Predictive test |
Full sequence |
Predictive test |
Full sequence |
Predictive test |
|
|
Method of gene detection (%) |
All ages |
- |
- |
26.73 |
73.27 |
30.05 |
69.95 |
22.81 |
77.19 |
|
|
1Female relatives only, including those already affected by breast and/or ovarian cancer at the time of the proband’s diagnosis. These include full sequence genetic tests and predictive tests unrelated to proband genetic testing or proband breast cancer diagnosis (e.g. another non-proband relative is found to carry a P/LP variant). For example, of the 1438 full sequence detections under the current practice scenario, 1181 (82%) were women unaffected at the time of the proband breast cancer diagnosis but who were later diagnosed with cancer, with the remainder being women tested due to a personal cancer diagnosis prior to the proband’s diagnosis 4The percentage of variants detected is calculated using the distribution of genotype totals above, for example for current practice, 1109/1504 BRCA1 (73.74%). Abbreviations: IQR, interquartile range; P/LP, pathogenic/likely pathogenic |
|
|||||||||
Comments 3: For the item "P/LP variant detected during lifetime," could you clarify the method used to calculate the percentage? Specifically, I would like to understand the denominator used for this calculation.
Response 3: The denominator was the total number of female relatives who were unaffected at the time of the proband’s breast cancer diagnosis, and who also carried a P/LP variant (presented in the rows directly above). We have clarified this in a table footnote as follows (see also Table 4 under response to comment 2 for footnote in context):
“4The percentage of variants detected is calculated using the distribution of genotype totals above, for example for current practice1109/1504 BRCA1 (73.74%). “
Comments 4: The same concern applies to Table 4: it is unclear how the percentages were calculated. Could you clarify the denominator used for these calculations? This information is essential for accurately interpreting the rates presented in the table.
Response 4: We agree this is unclear, as the denominator (total number of unaffected relatives with a P/LP variant) was presented only in the previous table. We have added a “total at risk” row to this table for clarification, and added “with a P/LP variant” to the row outcome description.
Comments 5: Regarding the items, it’s difficult to understand the distinctions between "All probands with a P/LP variant and their unaffected relatives," "Only relatives of probands with a P/LP variant," and "Relatives with a P/LP variant." Could you clarify the differences among these categories?
Response 5: We appreciate the wording of the different categories is confusing. To improve clarity, we have updated the categories in Table 4 (now Table 5) as follows:
- All women
- All probands with a P/LP variant, and their relatives
- Relatives only
- Relatives with a P/LP variant
We have added a footnote stating:
“Only female relatives unaffected at the time of the proband’s breast cancer diagnosis were included, with the exception of the “All women” category, which included all probands and all female relatives irrespective of P/LP variant carrier or cancer status.”
If it would be further helpful we could include the following venn diagram within the supplemental materials for clarity.
Figure. Venn diagram for categories in Table 5 life expectancy and life years saved estimates.

Round 2
Reviewer 1 Report
Comments and Suggestions for Authors
The manuscript can be accepted in the present form
Author Response
Thank you.
Reviewer 2 Report
Comments and Suggestions for Authors
Thank you for the significant effort you have put into your research. The topic is undoubtedly important, and the study has the potential to contribute valuable insights to the field. Your work demonstrates a thorough investigation and a commitment to advancing our understanding of this subject.
However, I found the presentation of results challenging to follow. Specifically, the terms used in the tables and the way the results are displayed lack clarity, making it difficult to interpret the data effectively.
To strengthen the impact of your work and make it more accessible to your audience, I recommend the following:
Consider reorganizing the tables to ensure that key information is presented in a logical and intuitive manner. Grouping related data or using consistent formatting may help.
Revise the visual design of tables to ensure clarity. For example, avoid overcrowding tables with excessive information and use distinct formatting.
Provide additional explanation in the main text to guide readers through the key points of the tables and figures.
These improvements will enhance the readability and overall quality of your manuscript, ensuring that your findings are communicated effectively to a broader audience.
Comments on the Quality of English Language
N/A
Author Response
Thank you to the reviewer for their kind words. We appreciate the comments regarding the difficulty in following the results as they were previously shown. We have made the following changes to make the results more accessible:
- Section 3.2.2. We have clarified the population of probands and relatives used to measure genetic testing outcomes
- Section 3.2.2, page 11 line 450. We have added explanation in the main text for the number of probands with breast cancer who underwent genetic testing.
Section 3.2.2 now reads:
3.2.2. Genetic testing outcomes
One million probands were simulated for each of the four genetic testing scenarios (Section 2.5.1), with approximately 120,000 of these probands diagnosed with breast cancer over their lifetime. Genetic testing outcomes are reported for the probands with breast cancer as described in Section 2.5.2, and their first- and second-degree relatives; Table 3 describes the underlying prevalence of P/LP variants in these probands and genetic testing rates for both probands and their relatives. Increased genetic testing rates were modelled to be implemented across the entire simulated population and not confined to probands, meaning the underlying prevalence of P/LP variants in cancer-affected probands decreased with each intervention from 3.53% for the no testing scenario to 2.39% for testing all breast cancers.
Under scenario 2 (current practice), 9279/120700 (7.69%) of probands underwent diagnostic genetic testing following their breast cancer diagnosis. This increased to 13.32% under the optimised referral scenario, and 99.99% for scenario 4, testing all breast cancers. There was a small number of probands (0.44-0.60%) who had a predictive test for a known family variant in all scenarios except for scenario 1 (no genetic testing). Each expansion of genetic testing led to an incremental decrease in secondary ovarian cancer incidence in probands with a P/LP variant from 557/4245 (13.12%) under no genetic testing, to 49/2867 (1.71%) under testing all breast cancers.
Probands with a P/LP variant had an average of 15.12-15.63 male and female relatives alive (including cancer-affected) at the time of the proband’s breast cancer diagnosis. There was a substantial increase in both diagnostic sequencing and predictive testing performed in relatives for each expanded testing scenario (Table 3).
The estimated mean number of at-risk female relatives with a P/LP variant who were unaffected by cancer at the time of the proband’s diagnosis ranged from 2.04-2.08 per proband carrier (including relatives aged under 18 years). Probands with PALB2 P/LP variants had more at-risk relatives on average at 2.10-2.22, compared to proband BRCA1 carriers at 1.74-1.90 relatives per proband and BRCA2 at 1.70-1.87. The estimated age distribution of unaffected female relatives tended to become slightly older with each scenario as genetic testing was expanded. Further detail regarding genetic outcomes by age-group is provided in Supplemental Table S11.
Current practice detected 39% of probands with a P/LP variant following their breast cancer diagnosis, compared with 50% for optimised referral, and 99% for testing all breast cancers (Figure 2). The proportion of unaffected female relatives with a P/LP variant who were identified through genetic testing was estimated to be 42% for current practice, 55% for optimised referral, and 85% for testing all breast cancers. For all scenarios, the majority (70-77%) of P/LP variants in relatives were found through predictive testing. The remaining 23-30% of carriers were only identified after they developed cancer in the years following the proband’s diagnosis. PALB2 carriers had the greatest increase in P/LP variant detection rates through expanded testing, starting at 21% for current practice, 34% for optimised referral, and 80% for testing all breast cancer. By contrast, BRCA1 P/LP variant detection increased from 74% to 91%, and BRCA2 from 45% to 87% for current practice and testing all breast cancers, respectively.
- Tables 3 and 4. These tables have been combined into a single table (Table 3. Genetic outcomes in probands diagnosed with breast cancer and their relatives). The data have been formatted in a more intuitive way, arranged by population subgroup. Information regarding the number of P/LP variant carriers and rates of P/LP variant detection under each scenario were moved instead to a bar plot (Figure 2).
Table 3. Genetic outcomes in probands diagnosed with breast cancer and their relatives
|
Group |
Outcome |
Scenario 1: No genetic testing
|
Scenario 2: Current practice
|
Scenario 3: Optimised referral of breast and ovarian cancer |
Scenario 4: Genetic testing for all breast cancers |
||||||
|
Probands (women diagnosed with breast cancer before age 80 years) |
Total number |
121 343 |
120 700 |
121 299 |
120 166 |
|
|||||
|
Age at diagnosis, mean (sd) |
60.53 (10.93) |
60.5 (10.94) |
60.47 (10.92) |
60.60 (10.90) |
|
||||||
|
Prevalence of P/LP variants, n (%) |
BRCA1 BRCA2 PALB2 Total |
990 (0.82%) 1908 (1.58%) 1347 (1.11%) 4245 (3.53%) |
787 (0.65%) 1605 (1.33%) 1278 (1.05%) 3670 (3.05%) |
714 (0.59%) 1539 (1.28%) 1291 (1.06%) 3544 (2.95%) |
684 (0.56%) 1235 (1.02%) 948 (0.78%) 2867 (2.39%) |
|
|||||
|
Genetic testing, diagnostic sequencing, n (%) |
0 (0%) |
9279 (7.69%) |
16 159 (13.32%) |
120 162 (99.99%) |
|
||||||
|
Genetic testing, predictive test, n (%) |
0 0%) |
537 (0.44%) |
727 (0.60%) |
664 (0.55%) |
|
||||||
|
Secondary ovarian cancer in P/LP variant carriers, n (%) |
557 (13.12%) |
283 (7.71%) |
200 (5.64%) |
49 (1.71%) |
|
||||||
|
Male and female relatives of probands who carry a P/LP variant |
Total number |
66 358 |
56 672 |
54 020 |
43 352 |
|
|||||
|
Average per proband |
15.63 |
15.44 |
15.24 |
15.12 |
|
||||||
|
Genetic testing, diagnostic sequencing, n (%)1 |
0 (%) |
1438 (2.54%) |
2238 (4.14%) |
3938 (9.08%) |
|
||||||
|
Genetic testing, predictive test, n (%)1 |
0 (0%) |
10 813 (19.09%) |
13 705 (25.37%) |
19 621 (45.26%) |
|
||||||
|
All cancer-unaffected female relatives, including non-carriers |
Total number |
30 736 |
26 661 |
25 319 |
20 461 |
|
|||||
|
Average per proband |
7.24 |
7.26 |
7.14 |
7.14 |
|
||||||
|
Age, median (IQR) |
34 (18, 58) |
34 (18, 59) |
34 (17, 59) |
34 (17, 60) |
|
||||||
|
Cancer-unaffected female relatives, excluding non-carriers |
Total number |
8881 |
7633 |
7217 |
5881 |
|
|||||
|
Average per proband |
2.08 |
2.07 |
2.06 |
2.04 |
|
||||||
|
Age, median (IQR) |
31 (17, 48) |
31 (17, 49) |
31 (17, 50) |
31 (17, 51) |
|
||||||
|
|
|
|
|
|
|
|
|
|
|
|
|
|
1These include diagnostic sequencing genetic tests and predictive tests unrelated to proband genetic testing or proband breast cancer diagnosis (e.g. another non-proband relative is found to carry a P/LP variant). For example, of the 1438 diagnostic sequencing detections under the current practice scenario, 1181 (82%) were women unaffected at the time of the proband breast cancer diagnosis but who were later diagnosed with cancer, with the remainder being women tested due to a personal cancer diagnosis prior to the proband’s diagnosis. Abbreviations: IQR, interquartile range; P/LP, pathogenic/likely pathogenic |
|
||||||||||
Figure 2. P/LP variant detection by scenario and gene. Relatives were female relatives with a P/LP variant, unaffected by cancer at the time of the proband’s breast cancer diagnosis.

- Table 5 (now Table 4). We have limited this outcomes table to two groups: (1) probands with a P/LP and their unaffected female relatives, and (2) unaffected female relatives with a P/LP variant. The outcomes related to cancer incidence in relatives were removed, as these are visually presented already in a figure (Figure 4). Risk-reducing surgery uptake was also moved to a figure instead (new Figure 3), so this table is focused on life expectancy and life years saved only. Section 3.2.3 now reads:
3.2.3. Cancer incidence and mortality
Increasing genetic referral and expanded testing resulted in higher estimated rates of risk-reducing surgery (Figure 4), and lower incidence of breast and ovarian cancer in relatives of probands with a P/LP variant in BRCA1, BRCA2 or PALB2 (Figure 4).
The reduced cancer incidence translated to an improvement in overall estimated life expectancy and reduction in the number of cancer-specific deaths (Figure 6). For probands with a P/LP variant and their unaffected female relatives, the average life expectancy increased by 0.122 years from optimised referral of all breast and ovarian cancer cases compared to current practice. Life years saved were estimated from the timepoint of the proband’s breast cancer diagnosis (the start of the genetic testing intervention). For the same group, additional life years saved through optimised referral were 0.127 compared to current practice. When comparing testing all breast cancer to optimised referral, although there was an increase in average life expectancy (0.126 years), there was a decrease in life years saved of 0.231. A similar pattern was seen when limited to only relatives with a P/LP variant (Table 4).

Figure 3. Risk-reducing surgery uptake in relatives with a P/LP variant. Relatives were female relatives with a P/LP variant, unaffected by cancer at the time of the proband’s breast cancer diagnosis. Abbreviations: BRRM, bilateral risk-reducing mastectomy; RRSO, risk-reducing salpingo-oophorectomy.
Table 4. Life expectancy and life years saved by genetic testing scenario.
|
Group |
Outcome |
No genetic testing |
Current practice |
Optimised referral of breast and ovarian cancer |
Genetic testing all breast cancers |
||||
|
|
|
Mean |
95% CI |
Mean |
95% CI |
Mean |
95% CI |
Mean |
95% CI |
|
Probands with a P/LP and their relatives1 |
Life expectancy |
84.672 |
(84.538, 84.806) |
85.067 |
(84.927, 85.208) |
85.189 |
(85.045, 85.333) |
85.315 |
(85.155, 85.475) |
|
Life years saved |
41.903 |
(41.606, 42.199) |
41.991 |
(41.671, 42.311) |
42.118 |
(41.787, 42.449) |
41.887 |
(41.519, 42.255) |
|
|
Relatives1 with a P/LP variant only |
Life expectancy |
82.304 |
(82.008, 82.600) |
83.443 |
(83.141, 83.746) |
83.973 |
(83.667, 84.279) |
84.159 |
(83.818, 84.501) |
|
Life years saved |
47.860 |
(47.373, 48.347) |
48.780 |
(48.241, 49.319) |
49.150 |
(48.595, 49.705) |
49.112 |
(48.495, 49.730) |
|
|
1Included relatives were female, and unaffected by cancer at the time of the proband’s breast cancer diagnosis. Abbreviations: CI, confidence interval; P/LP, pathogenic/likely pathogenic |
|||||||||
- Section 3.2.4 Sensitivity analysis, page 16 line 560. Additional explanation is provided regarding the impact of the assumptions introduced for the sensitivity analysis. A key outcome was the change in P/LP detection rates, which is now emphasised and highlighted in a new figure (Figure 6). Section 3.2.4 now reads:
3.2.4. Sensitivity analysis
NEEMO estimated similar numbers to a previous study for eligible at-risk female relatives, being an average of 1.52-1.55 per proband in the current study compared to 1.41-1.46 relatives per proband.22 The estimated additional increase in life expectancy of 374 days (undiscounted) for P/LP variant carriers was also similar to the previous study (289-419 days). Removing background diagnostic genetic testing in relatives combined with assuming 100% uptake of predictive testing led to scenario 4 (genetic testing of all breast cancers) generating a gain in life years saved for both groups (Supplemental Table S12).
The underlying prevalence of P/LP variants in probands with breast cancer remained constant across all scenarios at 3.49-3.56%. There was a substantial difference in P/LP variant detection rates when compared with the original NEEMO outcomes shown in Figure 2. For example, while only 13.9% of BRCA1 P/LP variants remained undetected for the original scenario 3 (optimised referral), this increased to 55.6% in the sensitivity analysis (Figure 6). For the original scenario 4 (testing all breast cancers) 14% P/LP variants in the relatives were never detected, compared to only 1% in the sensitivity analysis. The consequence of these changes was a much greater incremental difference in detection rates for relatives between scenarios 3 and 4 in the sensitivity analysis.

Figure 6. P/LP variant detection by scenario in the sensitivity analysis. Diagnostic genetic testing in relatives was removed for both the optimised referral and testing all breast cancers scenarios for the sensitivity analysis. The model also assumed 100% uptake of predictive testing in relatives of probands found to carry a P/LP variant. The no testing and current practice scenarios are shown for comparison but were not altered for the sensitivity analysis. Relatives were female relatives with a P/LP variant, unaffected by cancer at the time of the proband’s breast cancer diagnosis. Abbreviations: BC, breast cancer; P/LP, pathogenic/likely pathogenic.
- Supplementary materials, Table S12. This table was updated to reflect the new formatting of Table 4 in the main text (but retaining breast cancer incidence and risk-reducing surgery uptake).
|
Group |
Outcome |
Scenario 3: Optimised referral of breast and ovarian cancer |
Scenario 4: Genetic testing all breast cancers |
||
|
|
|
Mean |
95% CI |
Mean |
95% CI |
|
Probands with a P/LP and their relatives1 |
Life expectancy |
85.652 |
(85.516, 85.789) |
85.958 |
(85.825, 86.091) |
|
Life years saved |
42.118 |
(41.787, 42.449) |
41.887 |
(41.519, 42.255) |
|
|
Relatives1 with a P/LP variant only |
Life expectancy |
83.973 |
(83.667, 84.279) |
84.159 |
(83.818, 84.501) |
|
Life years saved |
49.150 |
(48.595, 49.705) |
49.112 |
(48.495, 49.730) |
|
|
|
|
|
|
|
|
|
Bilateral risk-reducing mastectomy uptake, n (%) |
1181 (16.9%) |
2747 (39.6%) |
|||
|
Risk-reducing salpingo-oophorectomy uptake, n (%) |
2739 (39.2%) |
6166 (89.0%) |
|||
|
Breast cancer incidence, n (%) |
2777 (39.7%) |
2055 (29.7%) |
|||
|
Ovarian cancer incidence, n (%) |
485 (6.9%) |
141 (2.0%) |
|||
|
Full P/LP variant detection following a cancer diagnosis was limited to the probands only. Relatives of probands identified as P/LP variant carriers were assumed to undergo predictive testing within one year at a rate of 100%. 1Included relatives were female, and unaffected by cancer at the time of the proband’s breast cancer diagnosis.
Abbreviations: CI, confidence interval; P/LP, pathogenic/likely pathogenic
|
|||||
Other changes
- Section 3.2.1 Genetic testing rates. Clarified which model outcomes were used to validate the rate of genetic testing in different subgroups.
- Updated the term for referring to full gene sequencing following a cancer diagnosis to “diagnostic genetic testing” / “diagnostic sequencing” instead of “primary genetic testing”

Round 3
Reviewer 2 Report
Comments and Suggestions for Authors
I appreciate the authors' efforts in addressing the previous comments and enhancing the quality of the manuscript. The revised version shows significant improvement, particularly in the clarity of data presentation and overall organization. I believe the manuscript is nearly ready for publication and would recommend it for acceptance after addressing a few minor revisions primarily focused on formatting adjustments.
Author Response
Thank you to the reviewer for their suggestions. All necessary changes will be implemented in accordance with the journal's requirements by us together with Cancers' specialised layout team.